# Formation with Non-Collision Control Strategies for Second-Order Multi-Agent Systems

**DOI:** 10.3390/e25060904

**Published:** 2023-06-06

**Authors:** Eduardo Aranda-Bricaire, Jaime González-Sierra

**Affiliations:** 1Centro de Investigación y de Estudios Avanzados del Instituto Politécnico Nacional, Av. IPN 2508, San Pedro Zacatenco, Mexico City 07360, Mexico; earanda@cinvestav.mx; 2Unidad Profesional Interdisciplinaria de Ingeniería Campus Hidalgo, Instituto Politécnico Nacional, Carretera Pachuca—Actopan Kilómetro 1+500, Distrito de Educación, Salud, Ciencia, Tecnología e Innovación, San Agustín Tlaxiaca 42162, Mexico

**Keywords:** collision avoidance, second-order agents, multi-agent systems, formation control

## Abstract

This article tackles formation control with non-collision for a multi-agent system with second-order dynamics. The nested saturation approach is proposed to solve the well-known formation control problem, allowing us to delimit the acceleration and velocity of each agent. On the other hand, repulsive vector fields (RVFs) are developed to avoid collisions among the agents. For this purpose, a parameter depending on the distances and velocities among the agents is designed to scale the RVFs adequately. It is shown that when the agents are at risk of collision, the distances among them are always greater than the safety distance. Numerical simulations and a comparison with a repulsive potential function (RPF) illustrate the agents’ performance.

## 1. Introduction

The collision avoidance problem is an ancient issue in the study of the motion of several mobile robots. One of the oldest contributions is the pioneering work of Khatib [1]. He proposed adding a reactive control law based on an artificial vector field under the danger of collisions. In turn, he proposed this vector field to be the negative gradient of an artificial potential function. His ideas dominated the field over the years. However, a main drawback was underlying: since the reactive action could annul the motion control law, the robots could become trapped at undesired equilibria.

Several solutions were proposed to overcome this problem, including designing navigation functions and using algorithms based on artificial intelligence to make real-time decisions. The first approach carries the drawback of the previous knowledge of the obstacles, and for the second, the need for rigorous mathematical proofs of non-collision and convergence. A thorough discussion of these difficulties can be found in [2].

To the best of our knowledge, the first rigorous proof of non-collision for any number of agents was published in the series of papers [3,4,5]. The idea was innovative: rather than proposing the “repulsive” vector field as the gradient of a potential function, adding a “repulsive” force, which may not be the gradient of any scalar function. This new approach was proven to be sound and illustrated in several simulations, animations, and real-time experiments. These new ideas were applied to first-order systems. Furthermore, other works that have tackled collision avoidance problems with formation control for first-order agents are [3,4,5,6,7,8,9,10,11,12,13,14], where different approaches and techniques were designed, developed and implemented in numerical simulations or real-time experiments. In the same context, in [15], single-integrator quadcopters were considered together with a distributed model predictive control scheme based on consensus theory to tackle collision avoidance. An algorithm to avoid collision in a time-efficient manner for unmanned aerial vehicles modeled by kinematics was proposed in [16] through the collision cone approach.

This paper aims to enlarge these ideas to second-order systems. At first sight, this complication seems very simple and even worthless for study. However, the technical complications are major. Let us be very specific. For a first-order system, the velocities of the robots or agents present in the system can be instantaneously modified, thus evading collisions. For a second-order system, this is far from being the case: even if an agent detects that there is another one—or obstacle, for that matter—it cannot stop its motion instantaneously due to the inertia of the agent. In this sense, some works addressing collision-avoidance tasks for second-order agents are [17,18,19,20,21,22,23]. For instance, ref. [17] comprehensively reviewed different collision avoidance approaches for unmanned aerial vehicles. Similarly, in [18], the approach is based on a rotational force field and a repulsive force field to allow robots to avoid and escape complex obstacle shapes. The collision avoidance for the kinematics and dynamics of the differential-drive robot, in the presence of the wheel’s skidding and slipping, is addressed in [19] by employing the potential functions approach under directed topologies. A leader–follower formation tracking with collision avoidance for multiple robots based on the distance and angle between them is developed, firstly in [20] for non-holonomic mobile robots and then in [21] for underactuated surface vessels. At the same time, in [22,23], the collision avoidance for autonomous underwater vehicles is solved by using potential functions [22] and by combining adaptive laws and the barrier Lyapunov function [23].

In [24], neural network techniques are employed to design a formation control with collision avoidance for a class of second-order nonlinear multi-agent systems by combining artificial potential field methods and leader–follower formation methods. In contrast, in [25], Euler–Lagrange systems with uncertainties are considered for the formation and collision avoidance based on repulsive potential functions. Under communication constraints, the topic for formation tracking with collision avoidance for second-order multi-agent systems is dealt with in [26] by incorporating a potential function into the formation tracking algorithm. Similarly, the formation containment problem with collision avoidance for satellites is solved in [27]. Finally, in [28], the formation control problem without collisions for second-order multi-agent systems is addressed, where the RVFs approach is used for collision avoidance.

This paper attempts to address these technical difficulties in two ways: first, to propose a bounded control law based on nested saturation functions to solve the formation control problem, and second, to adapt the case of first-order agents’ collision avoidance to the scenario of second-order agents. In very plain words, we propose defining critical distances: the “detection distance” *D* and the “security distance” *d*. The reactive control law starts to operate when any agent detects another one within the neighborhood of the “detection distance”, and our reactive control law is updated with this information to appropriately scale the “repulsive” vector field to mathematically ensure that both agents will never cross the “security distance”.

The main contribution of this paper is avoiding repulsive gradient functions, which may lead to local minima or undesirable equilibria. Instead, we propose a new perspective, presenting RVFs which are not the gradient of any potential function. Mathematically, they are not integrable [29,30].

## 2. Problem Statement

Let $N=\left\{R_{1},\ldots ,R_{n}\right\}$ be a set of *n* agents with second-order dynamics: (1)$$\begin{array}{ccc} {\dot{z}}_{i} & = & v_{i},\;i=1,\ldots ,n, \end{array}$$(2)$$\begin{array}{ccc} {\dot{v}}_{i} & = & u_{i}, \end{array}$$
where $z_{i}={\left[\begin{array}{cc} x_{i} & y_{i} \end{array}\right]}^{⊤}\in R^{2}$ is the position in the plane of the *i*-th agent while $v_{i}={\left[\begin{array}{cc} v_{x_{i}} & v_{y_{i}} \end{array}\right]}^{⊤}\in R^{2}$ is the linear and lateral velocity, respectively, and $u_{i}={\left[\begin{array}{cc} u_{x_{i}} & u_{y_{i}} \end{array}\right]}^{⊤}\in R^{2}$ is the linear and lateral acceleration, respectively, which correspond to the control inputs. In matrix form, one has: (3)$$\begin{array}{ccc} \dot{z} & = & v, \end{array}$$(4)$$\begin{array}{ccc} \dot{v} & = & u, \end{array}$$
where $z={\left[\begin{array}{ccc} z_{1}^{⊤} & \ldots & z_{n}^{⊤} \end{array}\right]}^{⊤}\in R^{2n}$ and $u={\left[\begin{array}{ccc} u_{1}^{⊤} & \ldots & u_{n}^{⊤} \end{array}\right]}^{⊤}\in R^{2n}$.

**Definition** **1**(Formation graph [31,32])**.**
*A formation graph G={N,E,C} that describes the communication among the agents consists of the following:*
*1.* *A set of vertices N corresponding to the n agents in the system.**2.* *A set of edges $E=\{\left(R_{j},R_{i}\right)\in N\times N,j\neq i\}$, which denotes the agent $R_{i}$ receive information from $R_{j}$.**3.* *A set of labels $C=\{c_{ji}\in R^{2}|\left(R_{j},R_{i}\right)\in N\times N,j\neq i\}$, where $c_{ji}$ is a vector that specifies the relative position between agents $R_{j}$ and $R_{i}$.*

**Definition** **2**(Laplacian matrix [31,32])**.**
*Given a formation graph G, the Laplacian matrix associated to G is given by*
$$L\left(G\right)=\Delta -A_{d},$$
*where $\Delta =\mathrm{diag}\{n_{1},\ldots ,n_{n}\}$ is the degree matrix with $n_{i}=\mathrm{card}\left\{N_{i}\right\}$, $N_{i}\subset N$ is a subset that contains the agents that can be detected by agent $R_{i}$, and $A_{d}$ is the adjacency matrix of G defined as*
$$a_{ij}=\left\{\begin{array}{ccc} 1 & \;if\; & (R_{j},R_{i})\in E,\\ 0 & \;otherwise.\; \end{array}\right.$$

**Definition** **3**(Linear saturation [33])**.**
*Given two positive constants L, M with L≤M, a function σ:R→R is said to be a linear saturation for (L,M) if it is a continuous, non-decreasing function satisfying the following:*
*1.* *xσ(x)>0, for all x≠0;**2.* *σ(x)=x when |x|≤L;**3.* *|σ(x)|≤M, for all x∈R.*

**Problem Statement.** Design, if possible, a bounded control strategy such that the following is true:The second-order multi-agent system avoids collisions among them, i.e.,
$$∥z_{i}\left(t\right)-z_{j}\left(t\right)∥>d,\forall t\geq 0,i\neq j,$$
where *d* is the safety distance.The agents reach a desired geometric pattern, i.e.,
$$\lim_{t\to \infty }\left(z_{i}-z_{i}^{*}\right)=0,$$
where, given a communication graph G, the desired position of each agent is defined as
(5)$$z_{i}^{*}=\frac{1}{n_{i}}\sum_{j\in N_{i}}\left(z_{i}+c_{ji}\right),\;i=1,\ldots ,n,$$
with $c_{ji}={\left[\begin{array}{cc} c_{ji_{x}} & c_{ji_{y}} \end{array}\right]}^{⊤}\in R^{2}$ are constant vectors which specify the desired relative position among the agents.

**Assumption** **1.**
*In this work, it is assumed that the communication graph possesses a spanning tree, i.e., there is one agent from which all the other agents can be reached following the communication channels. This implies that the so-called Laplacian matrix has exactly one eigenvalue at the origin and all the others have a positive real part.*


Let us define the position error as ${\tilde{z}}_{i}=n_{i}\left(z_{i}-z_{i}^{*}\right)$; thus, in matrix form, one has
(6)$$\tilde{z}=\left(L⊗I_{2}\right)z-c,$$
where ⊗ denotes the Kronecker product, and $I_{2}$ is the identity matrix, while
(7)$$c=\left[\begin{array}{c} \sum_{j\in N_{1}}c_{j1}\\ \sum_{j\in N_{2}}c_{j2}\\ \vdots \\ \sum_{j\in N_{n}}c_{jn} \end{array}\right],$$
is a vector that contains the formation vectors.

## 3. Control Strategy

The control law is given by
(8)$$u_{i}=\gamma_{i}+\beta_{i},$$
or in matrix form
(9)u=γ+β,
where $\gamma ={\left[\begin{array}{ccc} \gamma_{1}^{⊤} & \ldots & \gamma_{n}^{⊤} \end{array}\right]}^{⊤}\in R^{2n}$ is the control law that accomplishes the formation control, and it is based on the nested saturation functions methodology [33], while $\beta ={\left[\begin{array}{ccc} \beta_{1}^{⊤} & \ldots & \beta_{n}^{⊤} \end{array}\right]}^{⊤}\in R^{2n}$ is the repulsive part, and it is based on the RVF approach.

### 3.1. Formation Control

Specifically, γ is defined as
(10)$$\gamma =-\sigma_{2}\left(\left(L⊗I_{2}\right)v+\sigma_{1}\left(\eta \tilde{z}+v\right)\right),$$
where $\sigma_{k}\left(\cdot \right)$, for k=1,2, is a column vector with each element being a linear saturation function (Definition 3) such that each element of vector $\sigma_{k}\left(\cdot \right)$ is bounded from above by $M_{k}$, with $M_{k}$ as a constant satisfying $2M_{1}<M_{2}$ and η=1[1/s] as a constant that allows us to add the position error and the velocity.

**Remark** **1.***With the nested saturation approach, the formation control strategy given in* (Equation 10) *is bounded. In this sense, the acceleration of each agent is bounded by the constant $M_{2}$, and the velocity of each agent is bounded by the constant $M_{1}$.*

**Theorem** **1.***Let the control u=γ with γ defined in* (Equation 10) *be applied in system* (Equation 3)–(4)*; therefore, the position error $\tilde{z}$ and the velocity v asymptotically converge to zero, that is, $\lim_{t\to \infty }\tilde{z}=0$ and $\lim_{t\to \infty }v=0$. This means that the agents reach the desired geometric pattern.*

**Proof.** Let us consider the following change of coordinates $\xi_{1}=\eta \tilde{z}+v$ and $\xi_{2}=v$. The dynamics of these variables in the closed loop with control (Equation 10) is given by:
$$\begin{array}{ccc} {\dot{\xi }}_{1} & = & \left(L⊗I_{2}\right)\xi_{2}-\sigma_{2}\left(\left(L⊗I_{2}\right)\xi_{2}+\sigma_{1}\left(\xi_{1}\right)\right),\\ {\dot{\xi }}_{2} & = & -\sigma_{2}\left(\left(L⊗I_{2}\right)\xi_{2}+\sigma_{1}\left(\xi_{1}\right)\right). \end{array}$$According to the pioneering work of Teel [33], after some finite time, the argument of every function $\sigma_{k}\left(\cdot \right)$ has entered the region where the function is linear. After this finite time, the closed-loop system has the following structure:
$$\begin{array}{ccc} {\dot{\xi }}_{1} & = & -\xi_{1},\\ {\dot{\xi }}_{2} & = & -\left(L⊗I_{2}\right)\xi_{2}-\xi_{1}. \end{array}$$In matrix form, one has
(11)$$\dot{\xi }=A\xi ,$$
where $\xi ={\left[\begin{array}{cc} \xi_{1}^{⊤} & \xi_{2}^{⊤} \end{array}\right]}^{⊤}\in R^{4n}$ and
$$A=-\left[\begin{array}{cc} I_{2n} & 0_{2n}\\ I_{2n} & \left(L⊗I_{2}\right) \end{array}\right],$$
where $I_{2n}$ is the 2n×2n identity matrix and $0_{2n}$ is the 2n×2n zero matrix. Note that matrix *A* is a lower block triangular matrix; therefore, the eigenvalues of *A* are given by the eigenvalues of the diagonal block matrix, i.e., the eigenvalues of matrix $-I_{2n}$ and $-(L⊗I_{2})$. Since the communication graph possesses a spanning tree, its Laplacian L has n−1 eigenvalues with negative real part and one at the origin [32]. A simple application of the closed formation condition shows that all errors and velocities converge asymptotically to zero, i.e., $\lim_{t\to \infty }\tilde{z}=0$ and $\lim_{t\to \infty }v=0$, and the agents reach the desired geometric pattern.  □

### 3.2. Collision Avoidance Control

Before proceeding, the following standing assumptions are made:

**Assumption** **2.**
*The magnitude of the formation vectors is greater than the sensing distance D and the safety distance d, i.e., $∥c_{ij}∥>D>d$, ∀i≠j. Therefore, the desired position of each agent is located outside the sensing distance.*


**Assumption** **3.**
*The initial conditions of all the agents satisfy $∥z_{i}\left(0\right)-z_{j}\left(0\right)∥>D$, ∀i≠j. Thus, the initial conditions of the agents are outside the sensing distance.*


The RVF is defined as
(12)$$\beta_{i}=-\epsilon \sum_{j\in M_{i}}\delta_{ij}\left[\begin{array}{c} \left(x_{j}-x_{i}\right)-\left(y_{j}-y_{i}\right)\\ \left(x_{j}-x_{i}\right)+\left(y_{j}-y_{i}\right) \end{array}\right],$$
where $M_{i}$ is a set composed of all those agents that are at risk of collision with agent $R_{i}$, i.e.,
$$M_{i}=\left\{R_{j}\in N|∥z_{i}-z_{j}∥\leq D\right\},\;i=1,\cdots ,n,$$
ϵ is a design parameter to be defined later, *D* is the sensing distance, and $\delta_{ij}$ is a function that turns on and turns off the RVF, defined as
(13)$$\begin{array}{ccc} \delta_{ij} & = & \left\{\begin{array}{ccc} 1 & \;\mathrm{if}\; & d_{ij}\leq D,\\ 0 & \;\mathrm{if}\; & d_{ij}>D, \end{array}\right. \end{array}$$
with $d_{ij}=∥z_{i}-z_{j}∥$ as the distance between two agents. For designing the parameter ϵ, the relative distance and relative velocity variables between two agents are defined as: (14)$$\begin{array}{cccc} p_{ij} & =x_{j}-x_{i},\;\;\; & q_{ij} & =y_{j}-y_{i}, \end{array}$$(15)$$\begin{array}{cccc} r_{ij} & ={\dot{x}}_{j}-{\dot{x}}_{i},\;\;\; & s_{ij} & ={\dot{y}}_{j}-{\dot{y}}_{i}. \end{array}$$

Furthermore, the following surface is proposed:(16)$$S_{ij}=p_{ij}^{2}+q_{ij}^{2}-d^{2}>0,$$
where d<D is the safety distance. Under this scenario, the trajectories defined by $p_{ij}$ and $q_{ij}$ must lie outside the region $S_{ij}>0$. Therefore, one needs to verify that the surface $S_{ij}=0$ is repulsive from the outer region.

In the first step, the scaling factor ϵ will be designed, taking into account the simplest case, that is, two agents are at risk of collision, i.e., $N=\left\{R_{a},R_{b}\right\}$.

**Proposition** **1.**
*The relative velocities are bounded from above as*

(17)
$$\begin{array}{cccc} |r_{ij}| & \leq 2{\bar{M}}_{1},\;\;\; & |s_{ij}| & \leq 2{\bar{M}}_{1}, \end{array}$$

*where ${\bar{M}}_{1}=M_{1}+\alpha$ and α>0 is a constant that comes from the velocity evasion maneuver, while the vector*

(18)
$$∥\left[\begin{array}{cc} r_{ij} & s_{ij} \end{array}\right]∥\leq 2\sqrt{2}{\bar{M}}_{1}.$$



**Proof.** Note that (15) has the following bound:
$$\begin{array}{cccc} |r_{ij}| & \leq |{\dot{x}}_{j}\left|+\right|{\dot{x}}_{i}|,\;\;\; & |s_{ij}| & \leq |{\dot{y}}_{j}\left|+\right|{\dot{y}}_{i}|. \end{array}$$Recall that the velocity of each agent is bounded by $M_{1}$. Nevertheless, this bound is given by considering only the attractive part of the control strategy (Equation 8). When taking into account the attractive and repulsive terms of (Equation 8), the velocity of each agent will be bounded by $|{\dot{x}}_{j}|\leq M_{1}+\alpha ={\bar{M}}_{1}$ with α>0. Therefore, the relative velocities are bounded as
$$\begin{array}{cccc} |r_{ij}| & \leq 2{\bar{M}}_{1},\;\;\; & |s_{ij}| & \leq 2{\bar{M}}_{1}. \end{array}$$Knowing the bounds of the relative velocities, it is straightforward that $∥\left[\begin{array}{cc} r_{ij} & s_{ij} \end{array}\right]∥\leq 2\sqrt{2}{\bar{M}}_{1}$.  □

**Theorem** **2.***Let the control* (Equation 8) *be applied in the system* (Equation 1)–(2) *along the Definitions* (Equation 10)*,* (Equation 12) *and* (Equation 13)*. It is assumed that at time $t_{0}$, there is an agent, $R_{a}$, at risk of collision with agent $R_{b}$, that is $d_{ij}\leq D$; thus, $\delta_{ab}=1$. If*
$$\epsilon >\frac{1}{d^{2}}\left(\frac{4{\bar{M}}_{1}^{2}D^{2}}{D^{2}-d^{2}}+\sqrt{2}DM_{2}\right),$$
*the agents will avoid collisions, that is $∥z_{a}\left(t\right)-z_{b}\left(t\right)∥>d$ for all $t>t_{0}$.*

**Proof.** The dynamics of (Equation 14)–(15) is given as follows:
$$\begin{array}{ccc} \left[\begin{array}{c} {\dot{p}}_{ab}\\ {\dot{q}}_{ab} \end{array}\right] & = & \left[\begin{array}{c} r_{ab}\\ s_{ab} \end{array}\right],\\ \left[\begin{array}{c} {\dot{r}}_{ab}\\ {\dot{s}}_{ab} \end{array}\right] & = & -\sigma_{2_{b}}\left(v_{b}+\sigma_{1_{b}}\left(\eta {\tilde{z}}_{b}+v_{b}\right)\right)+\sigma_{2_{a}}\left(v_{a}+\sigma_{1_{a}}\left(\eta {\tilde{z}}_{a}+v_{a}\right)\right)+2\epsilon \delta_{ab}\left[\begin{array}{c} p_{ab}-q_{ab}\\ p_{ab}+q_{ab} \end{array}\right]. \end{array}$$On the other hand, the dynamics of the surface (Equation 16) is given by:
(19)$$\begin{array}{ccc} {\dot{S}}_{ab} & = & 2\left[\begin{array}{cc} p_{ab} & q_{ab} \end{array}\right]\left[\begin{array}{c} r_{ab}\\ s_{ab} \end{array}\right], \end{array}$$
(20)$$\begin{array}{ccc} {\ddot{S}}_{ab} & = & 2\left(r_{ab}^{2}+s_{ab}^{2}\right)-2\left[\begin{array}{cc} p_{ab} & q_{ab} \end{array}\right]\left[\sigma_{2_{b}}\left(v_{b}+\sigma_{1_{b}}\left(\eta {\tilde{z}}_{b}+v_{b}\right)\right)-\sigma_{2_{a}}\left(v_{a}+\sigma_{1_{a}}\left(\eta {\tilde{z}}_{a}+v_{a}\right)\right)\right]\\ & & +4\epsilon \delta_{ab}\left[\begin{array}{cc} p_{ab} & q_{ab} \end{array}\right]\left[\begin{array}{c} p_{ab}-q_{ab}\\ p_{ab}+q_{ab} \end{array}\right]. \end{array}$$Applying the norm operator, one has
$$\begin{array}{c} ∥\sigma_{2_{b}}\left(\cdot \right)-\sigma_{2_{a}}\left(\cdot \right)∥\leq ∥\sigma_{2_{b}}\left(\cdot \right)∥+∥\sigma_{2_{a}}\left(\cdot \right)∥, \end{array}$$
and recall that each element of $\sigma_{k}\left(\cdot \right)$ is bounded by $M_{2}$; therefore,
$$\begin{array}{c} ∥\sigma_{2_{b}}\left(\cdot \right)-\sigma_{2_{a}}\left(\cdot \right)∥\leq 2\sqrt{2}M_{2},\\ -∥\sigma_{2_{b}}\left(\cdot \right)-\sigma_{2_{a}}\left(\cdot \right)∥\geq -2\sqrt{2}M_{2}, \end{array}$$
and ${\ddot{S}}_{ab}$ can be bounded from below as
(21)$${\dot{S}}_{ab}\geq -4\sqrt{2}M_{2}∥\left[\begin{array}{cc} p_{ab} & q_{ab} \end{array}\right]∥+4\epsilon \left(p_{ab}^{2}+q_{ab}^{2}\right).$$Note that the term $r_{ab}^{2}+s_{ab}^{2}\geq 0$; therefore, it does not appear in (Equation 21). Meanwhile, the term $-∥\left[\begin{array}{cc} p_{ab} & q_{ab} \end{array}\right]∥\geq -D$ and $p_{ab}^{2}+q_{ab}^{2}\geq d^{2}$, and hence, (Equation 21) is bounded from below by
(22)$${\ddot{S}}_{ab}\geq -4\sqrt{2}DM_{2}+4\epsilon d^{2}>0.$$Integrating (Equation 22) with integration limits $t_{0}$ and *t*, one has
(23)$$\begin{array}{c} {\dot{S}}_{ab}\left(t\right)\geq -4\left(\sqrt{2}DM_{2}-\epsilon d^{2}\right)\left(t-t_{0}\right)+{\dot{S}}_{ab}\left(t_{0}\right)>0. \end{array}$$Since the agents are approaching each other, ${\dot{S}}_{ab}\left(t_{0}\right)=-\left|{\dot{S}}_{ab}\left(t_{0}\right)\right|$, and (Equation 23) is rewritten as
(24)$$\begin{array}{c} {\dot{S}}_{ab}\left(t\right)\geq -4\left(\sqrt{2}DM_{2}-\epsilon d^{2}\right)\left(t-t_{0}\right)-\left|{\dot{S}}_{ab}\left(t_{0}\right)\right|>0. \end{array}$$Note that from (Equation 19) and (Equation 18), ${\dot{S}}_{ab}$ is bounded by
$${\dot{S}}_{ab}\leq 4D\sqrt{2}{\bar{M}}_{1},$$
and (Equation 24) is rewritten as
(25)$${\dot{S}}_{ab}\left(t\right)\geq -4\left(\sqrt{2}DM_{2}-\epsilon d^{2}\right)\left(t-t_{0}\right)-4D\sqrt{2}{\bar{M}}_{1}>0.$$Integrating (Equation 25) with integration limits $t_{0}$ and *t*, one has
$$\begin{array}{c} S_{ab}\left(t\right)\geq -2\left(\sqrt{2}DM_{2}-\epsilon d^{2}\right){\left(t-t_{0}\right)}^{2}-4D\sqrt{2}{\bar{M}}_{1}\left(t-t_{0}\right)+S_{ab}\left(t_{0}\right). \end{array}$$Taking into account that the analysis is performed when the agents enter the sensing zone, we have $S_{ab}\left(t_{0}\right)=D^{2}-d^{2}$, and $S_{ab}\left(t\right)$ is rewritten as
(26)$$\begin{array}{c} S_{ab}\left(t\right)\geq -2\left(\sqrt{2}DM_{2}-\epsilon d^{2}\right){\left(t-t_{0}\right)}^{2}-4D\sqrt{2}{\bar{M}}_{1}\left(t-t_{0}\right)+D^{2}-d^{2}>0. \end{array}$$Note that (Equation 26) depends on the term $t-t_{0}$; therefore, (Equation 25) is used to clear $t-t_{0}$, obtaining the following expression:
(27)$$\begin{array}{c} t-t_{0}=\frac{\sqrt{2}{\bar{M}}_{1}D}{\epsilon d^{2}-\sqrt{2}DM_{2}}, \end{array}$$
and, substituting (Equation 27) in (Equation 26), one has
$$\begin{array}{c} S_{ab}\left(t\right)\geq D^{2}-d^{2}-\frac{4{\bar{M}}_{1}^{2}D^{2}}{\epsilon d^{2}-\sqrt{2}DM_{2}}>0. \end{array}$$From this last expression, it is possible to obtain the value of ϵ to scale the RVF, and it is given by
$$\epsilon >\frac{1}{d^{2}}\left(\frac{4{\bar{M}}_{1}^{2}D^{2}}{D^{2}-d^{2}}+\sqrt{2}DM_{2}\right).$$With this value, the RVFs are scaled adequately, and the agents avoid collisions, that is, $∥z_{a}\left(t\right)-z_{b}\left(t\right)∥>d$ for all $t>t_{0}$.  □

Based on the previous analysis, the following step is to determine the value of ϵ when an agent $R_{a}$ is at risk of collision with λ agents, for λ=1,…,n−1. In this sense, the following theorem states the main contribution of this work.

**Theorem** **3.***Let the control* (Equation 8) *be applied in the system* (Equation 1)–(2) *along the Definitions* (Equation 10)*,* (Equation 12) *and* (Equation 13)*. Defining $\rho =\mathrm{card}\left\{M_{a}\right\}$ as the number of agents that are in danger of collision with $R_{a}$, if*
$$\epsilon >\frac{2}{d^{2}\left(2+\left(\rho -1\right)\sqrt{2}\right)}\left(\frac{4{\bar{M}}_{1}^{2}D^{2}}{D^{2}-d^{2}}+\sqrt{2}DM_{2}\right).$$
*therefore, the agents will avoid collisions, that is $∥z_{a}\left(t\right)-z_{j}\left(t\right)∥>d$ for all $t>t_{0}$.*

**Proof.** The dynamics of (Equation 14)–(15) is given as follows:
$$\begin{array}{ccc} \left[\begin{array}{c} {\dot{p}}_{aj}\\ {\dot{q}}_{aj} \end{array}\right] & = & \left[\begin{array}{c} r_{aj}\\ s_{aj} \end{array}\right],\;j\in M_{a},\\ \left[\begin{array}{c} {\dot{r}}_{aj}\\ {\dot{s}}_{aj} \end{array}\right] & = & -\sigma_{2_{j}}\left(v_{j}+\sigma_{1_{j}}\left(\eta {\tilde{z}}_{j}+v_{j}\right)\right)+\sigma_{2_{a}}\left(v_{a}+\sigma_{1_{a}}\left(\eta {\tilde{z}}_{a}+v_{a}\right)\right)\\ & & +2\epsilon \delta_{aj}\left[\begin{array}{c} p_{aj}-q_{aj}\\ p_{aj}+q_{aj} \end{array}\right]+\epsilon \sum_{\lambda \in M_{a},\lambda \neq j}\delta_{a\lambda }\left[\begin{array}{c} p_{a\lambda }-q_{a\lambda }\\ p_{a\lambda }+q_{a\lambda } \end{array}\right],\;j\in M_{a}. \end{array}$$Then, a set of surfaces is defined as
$$S_{aj}=p_{aj}^{2}+q_{aj}^{2}-d^{2}>0,\;j\in M_{a}.$$Based on the results of Theorem 2, the second time derivative of each surface can be bounded as
$${\ddot{S}}_{aj}\geq -4\sqrt{2}DM_{2}+\epsilon d^{2}\left[4+2\left(\rho -1\right)\sqrt{2}\right]>0,$$
and, integrating with integration limits $t_{0}$ and *t*, one obtains the following expression:
(28)$${\dot{S}}_{aj}\left(t\right)\geq \left[\epsilon d^{2}\left(4+2\left(\rho -1\right)\sqrt{2}\right)-4\sqrt{2}DM_{2}\right]\left(t-t_{0}\right)-4D\sqrt{2}{\bar{M}}_{1}>0.$$Integrating (Equation 28), with integration limits $t_{0}$ and *t*, one has
(29)$$\begin{array}{c} S_{aj}\left(t\right)\geq \left[\epsilon d^{2}\left(2+\left(\rho -1\right)\sqrt{2}\right)-2\sqrt{2}DM_{2}\right]{\left(t-t_{0}\right)}^{2}-4D\sqrt{2}{\bar{M}}_{1}\left(t-t_{0}\right)>0. \end{array}$$From (Equation 28), one can obtain the following:
(30)$$\begin{array}{c} t-t_{0}=\frac{2\sqrt{2}{\bar{M}}_{1}D}{\epsilon d^{2}\left(2+\left(\rho -1\right)\sqrt{2}\right)-2\sqrt{2}DM_{2}}, \end{array}$$
then, substituting (Equation 30) in (Equation 29), one obtains
$$\begin{array}{c} S_{aj}\left(t\right)\geq D^{2}-d^{2}-\frac{8{\bar{M}}_{1}^{2}D^{2}}{\epsilon d^{2}\left(2+\left(\rho -1\right)\sqrt{2}\right)-2\sqrt{2}DM_{2}}>0. \end{array}$$From this last expression, it is possible to obtain the value of ϵ given by
$$\epsilon >\frac{2}{d^{2}\left(2+\left(\rho -1\right)\sqrt{2}\right)}\left(\frac{4{\bar{M}}_{1}^{2}D^{2}}{D^{2}-d^{2}}+\sqrt{2}DM_{2}\right).$$With this value, the RVFs are scaled adequately, and the agents avoid collisions, that is, $∥z_{a}\left(t\right)-z_{j}\left(t\right)∥>d$ for all $t>t_{0}$.  □

**Remark** **2.***Note that* (Equation 12) *can be rewritten in terms of the relative distance variables as*
$$\beta_{i}=-\epsilon \sum_{j\in M_{i}}\delta_{ij}\left[\begin{array}{c} p_{ij}-q_{ij}\\ p_{ij}+q_{ij} \end{array}\right].$$
*Taking into account that there is a risk of collision, thus $\delta_{ij}=1$ for all $j\in M_{i}$, and, applying the norm operator, one has*

$$∥\beta_{i}∥\leq \epsilon \sum_{j\in M_{i}}∥\left[\begin{array}{c} p_{ij}-q_{ij}\\ p_{ij}+q_{ij} \end{array}\right]∥.$$


*Furthermore, one has the following relation:*

$$∥\left[\begin{array}{c} p_{ij}-q_{ij}\\ p_{ij}+q_{ij} \end{array}\right]∥=\sqrt{{\left(p_{ij}-q_{ij}\right)}^{2}+{\left(p_{ij}+q_{ij}\right)}^{2}}=\sqrt{2(p_{ij}^{2}+q_{ij}^{2})}.$$


*Since $\delta_{ij}=1$, it means that $\sqrt{p_{ij}^{2}+q_{ij}^{2}}\leq D$, and considering $\rho =\mathrm{card}\left\{M_{a}\right\}$ as the number of agents that are in danger of collision with $R_{a}$, the bound of the RVFs is*

$$∥\beta_{i}∥\leq \sqrt{2}D\epsilon \rho .$$



Since (Equation 10) and (Equation 12) are bounded, therefore, the control strategy given in (Equation 8) is also bounded.

## 4. Numerical Simulations

In this section, two numerical simulations are presented to illustrate the performance of a group of agents when they are at risk of collision. Furthermore, to show the effectiveness of the RVFs approach, a comparison between the repulsive potential approach (RPA) reported in [34] by Tatsch and the RVFs is carried out. In this sense, following the methodology in [34], the control law can be bounded as
$$u_{i}=\left\{\begin{array}{ccc} \gamma_{i}+\beta_{i} & \mathrm{if} & ∥\gamma_{i}+\beta_{i}∥\leq a_{\max},\\ \frac{\gamma_{i}+\beta_{i}}{∥\gamma_{i}+\beta_{i}∥}a_{\max} & \mathrm{otherwise}, \end{array}\right.$$
where $a_{\max}$ is the maximum acceleration of each agent.

### 4.1. Two Agents

The first numerical simulation consists of the position interchange of two agents with a radius of 0.2 [m]. In this sense, the initial positions of the agents are $z_{1}\left(0\right)={\left[\begin{array}{cc} 3 & 0 \end{array}\right]}^{⊤}$ and $z_{2}\left(0\right)={\left[\begin{array}{cc} -3 & 0 \end{array}\right]}^{⊤}$; the communication topology is an undirected graph, where the formation vectors are given by $c_{21}={\left[\begin{array}{cc} -4 & 0 \end{array}\right]}^{⊤}$ and $c_{12}=-c_{21}$. The sensing distance is set to D=1 [m]; the safety distance is set to d=0.5 [m]; and the bounds of the saturation functions are $M_{1}=0.9$, $M_{2}=2$ and $a_{\max}=2$ [m/s${}^{2}$]. With these parameters, ϵ>28.5; therefore, one sets ϵ=29. Figure 1 depicts the trajectory comparison between the RVFs and the RPA of both agents in different time instances. Note that when using the RVFs approach, the agents are coming closer to each other; each agent enters the sensing distance; the RVFs activate; and the agents perform collision avoidance. Once the agents are outside the sensing distance, the formation control prevails, and the agents reach the desired formation. On the other hand, when using the RPA, it is possible to note that the agents fail in the simplest collision avoidance case, i.e., they get stuck into an undesired equilibrium point. Under this scenario, it is clear that the proposed methodology obtains better results than the one proposed by Tatsch [34].

Figure 2 illustrates the trajectory in the plane $p_{12}-q_{12}$, where the origin means that the agents collide. The circle in red is the sensing distance, while the magenta circle is the safety distance. When the trajectory enters the sensing distance, the RVFs are activated, and the collision is avoided.

Figure 3 presents the trajectory in the plane $S-\dot{S}$. Note that $\dot{S}$ is negative when the agents approach each other. When $S=D^{2}-d^{2}$, the RVFs are activated, and collision avoidance is performed.

The position error is shown in Figure 4, where it becomes evident that they converge to zero.

Finally, the control inputs required to perform the agents’ motion are illustrated in Figure 5. Note that when the RVFs are activated, the control inputs increase their magnitude (between 6 and 9 [s]), but they remain bounded, i.e., $u_{i}\in \left[-2,2\right]$.

### 4.2. Ten Agents with a Mixed Graph Communication Topology

This numerical simulation consists of 10 agents moving on a horizontal plane. The initial positions of the agents are $z_{1}\left(0\right)={\left[\begin{array}{cc} 5 & 0 \end{array}\right]}^{⊤}$, $z_{2}\left(0\right)={\left[\begin{array}{cc} -5 & 0 \end{array}\right]}^{⊤}$, $z_{3}\left(0\right)={\left[\begin{array}{cc} 3 & 0 \end{array}\right]}^{⊤}$, $z_{4}\left(0\right)={\left[\begin{array}{cc} -3 & 0 \end{array}\right]}^{⊤}$, $z_{5}\left(0\right)={\left[\begin{array}{cc} 1 & 0 \end{array}\right]}^{⊤}$, $z_{6}\left(0\right)={\left[\begin{array}{cc} -1 & 0 \end{array}\right]}^{⊤}$, $z_{7}\left(0\right)={\left[\begin{array}{cc} 0 & 5 \end{array}\right]}^{⊤}$, $z_{8}\left(0\right)={\left[\begin{array}{cc} 0 & -5 \end{array}\right]}^{⊤}$, $z_{9}\left(0\right)={\left[\begin{array}{cc} 0 & 3 \end{array}\right]}^{⊤}$ and $z_{10}\left(0\right)={\left[\begin{array}{cc} 0 & -3 \end{array}\right]}^{⊤}$; the communication topology is a mixed graph, where the formation vectors are given by $c_{32}=c_{45}=c_{56}={\left[\begin{array}{cc} 4 & 0 \end{array}\right]}^{⊤}$, $c_{83}=c_{78}=c_{610}=c_{109}=c_{91}={\left[\begin{array}{cc} 0 & 2 \end{array}\right]}^{⊤}$, $c_{12}=c_{23}=-c_{32}$, $c_{38}=c_{74}=-c_{83}$, $c_{87}=-c_{78}$, $c_{54}=-c_{45}$, $c_{19}=-c_{91}$, and $c_{106}=-c_{610}$. The sensing distance, the safety distance, and the bounds of the saturation functions are the same as in the previous simulation. With these parameters, ϵ>4.29; therefore, one sets ϵ=5. Figure 6 depicts the trajectory in the plane of the agents when using the RVFs approach. Once again, note that when the agents come closer to each other, each agent enters the sensing distance; the RVFs are activated; and the agents perform collision avoidance. Once the agents are outside the sensing distance, the formation control prevails, and the agents reach the desired formation.

On the other hand, Figure 7 illustrates the trajectory in the plane of the agents when using the RPA. It can be noticed that the agents’ motion is similar to the behavior of Figure 6. When the agents are at risk of collision, they perform the collision task, and when they are outside the sensing distance, the formation control prevails, and the agents achieve the desired formation.

The position errors are illustrated in Figure 8a for the RVFs approach, while Figure 8b presents the positions error for the RPA. In both cases, it is clear that their positions converge to zero, and therefore, the agents achieve the desired formation.

Figure 9a shows the distance among the agents when using the RVFs approach, while Figure 9b depicts the distance among the agents with the RPA. It is worth pointing out that when using the RPA, some of the distances among the agents are closer to the safety distance. In both cases, it is worth mentioning that when the agents are at risk of collision, the RVFs and the RPA activate, and the agents achieve collision avoidance. In this sense, the distance among the agents is greater than the safety distance d=0.5 [m].

Finally, Figure 10a presents the control inputs required to perform their motion with the RVF approach, while Figure 10b shows the control inputs required to perform their motion with the RPA. Note that when the agents are at risk of collision, the control input magnitude increases. Furthermore, when using the RVFs approach, chattering effects appear in the signals. This comes from the fact that the RVFs are discontinuous and, therefore, the RVFs are turning on and off constantly until the agents escape the sensing distance. In both cases, the control inputs remain bounded, i.e., $u_{i}\in \left[-2,2\right]$.

## 5. Conclusions

This article addresses formation control for a second-order multi-agent system, avoiding collisions among them. The nested saturation approach is proposed to accomplish formation control if the communication graph is strongly connected, while RVFs are designed to avoid collisions. Due to the formation control being bounded, obtaining a constant parameter to scale the RVFs adequately is possible. Theoretically and with numerical simulations, it is proved that when the agents are at risk of collision, the distances among them are greater than the safety distance. Furthermore, the proposed scaling of the RVF is general, independent of the number of agents involved. For future work, collision avoidance analysis in a heterogeneous multi-agent system will be considered, as well as a smooth function to turn on and off the RVFs to eliminate chattering in the control signals.

## Figures and Tables

**Figure 1 entropy-25-00904-f001:**
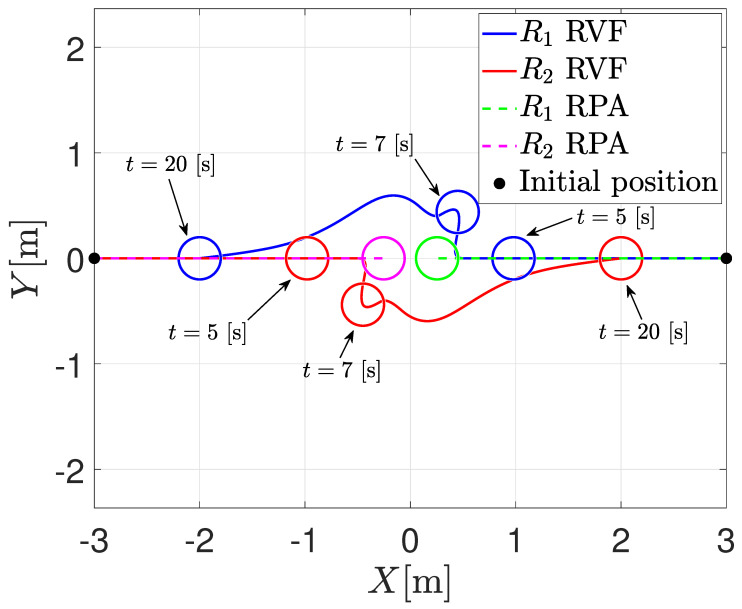
Trajectory in the plane for both agents.

**Figure 2 entropy-25-00904-f002:**
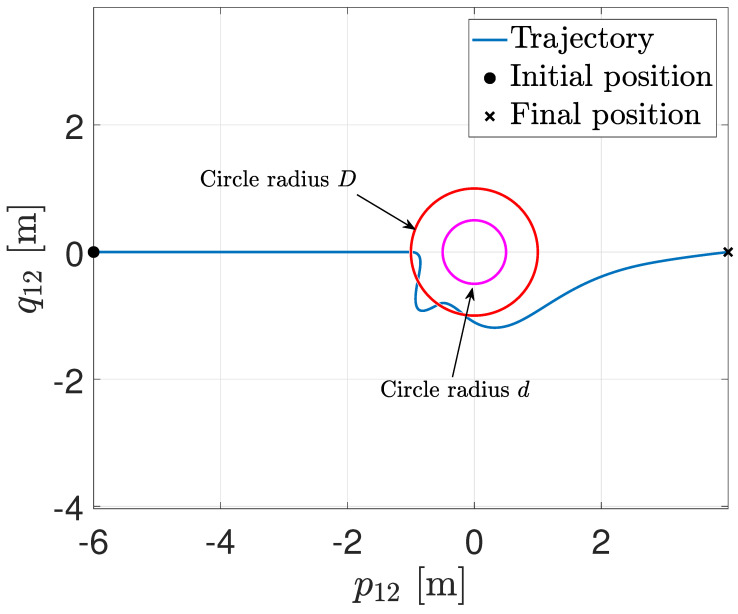
Trajectory in the plane $p_{ij}-q_{ij}$.

**Figure 3 entropy-25-00904-f003:**
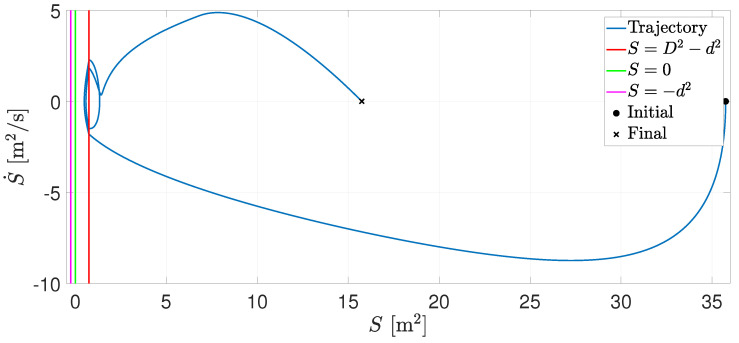
Trajectory in the plane $S-\dot{S}$.

**Figure 4 entropy-25-00904-f004:**
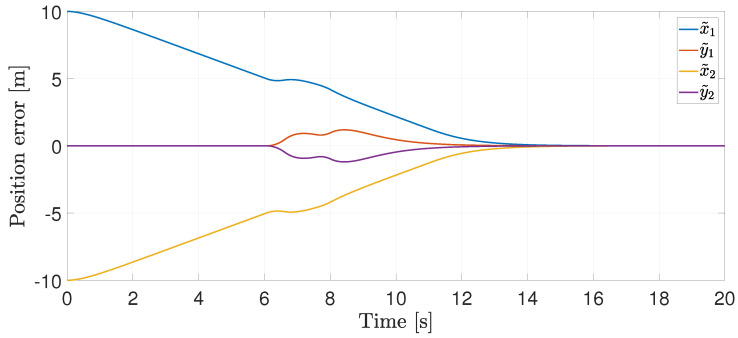
Position error.

**Figure 5 entropy-25-00904-f005:**
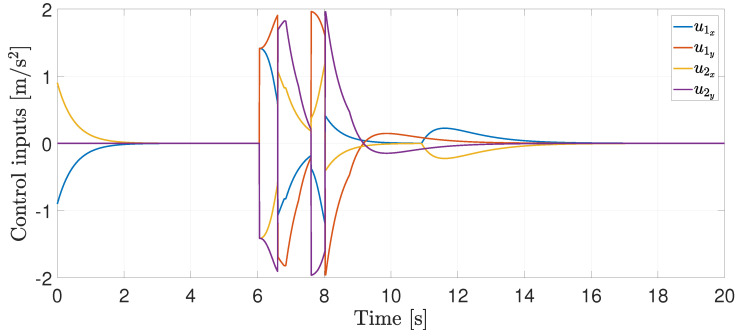
Control inputs.

**Figure 6 entropy-25-00904-f006:**
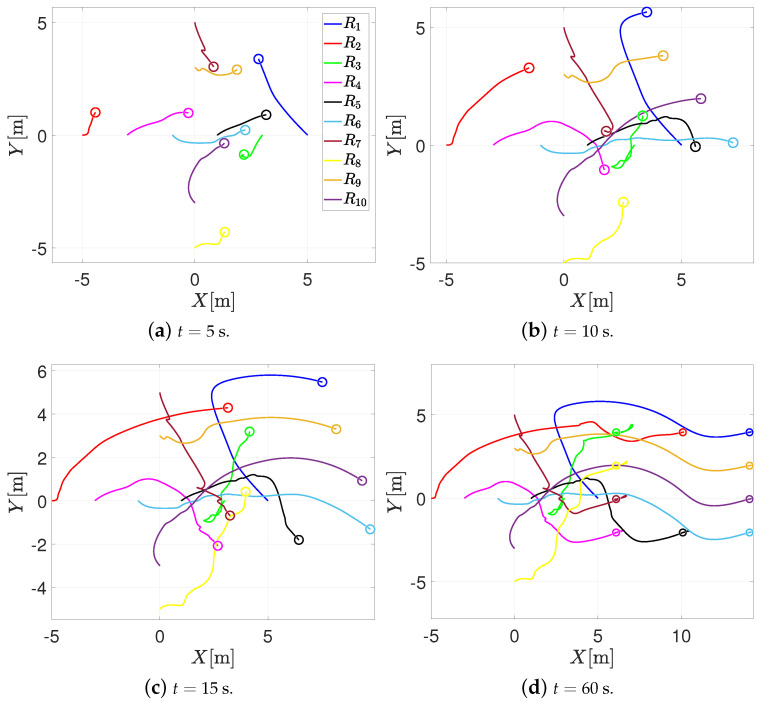
Trajectory in the plane of the agents with the RVF approach.

**Figure 7 entropy-25-00904-f007:**
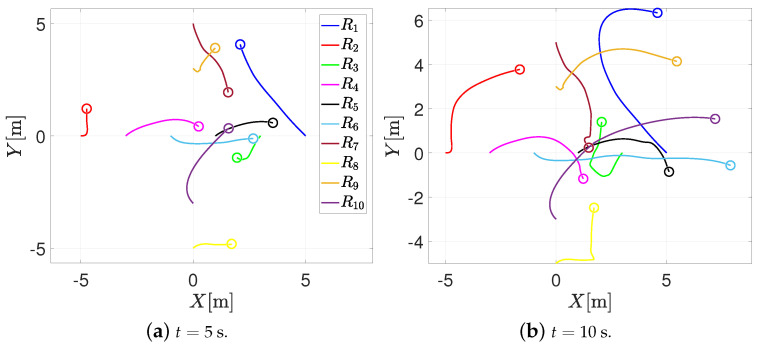
Trajectory in the plane of the agents with the RPA.

**Figure 8 entropy-25-00904-f008:**
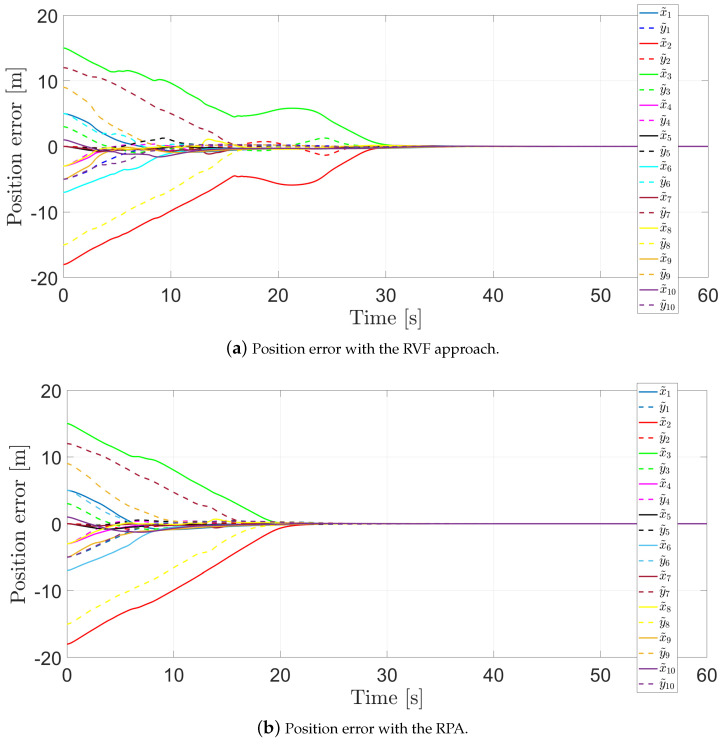
Position error.

**Figure 9 entropy-25-00904-f009:**
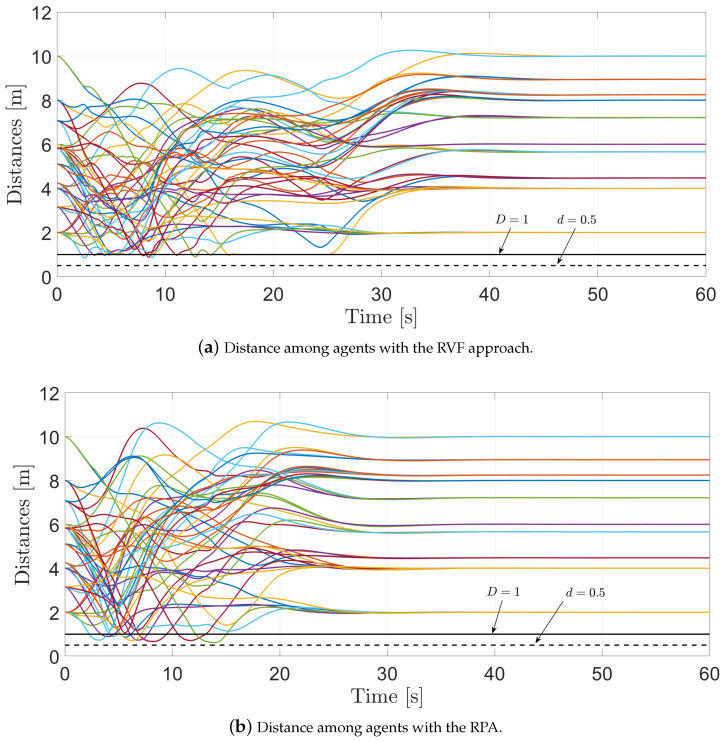
Distances among the agents.

**Figure 10 entropy-25-00904-f010:**
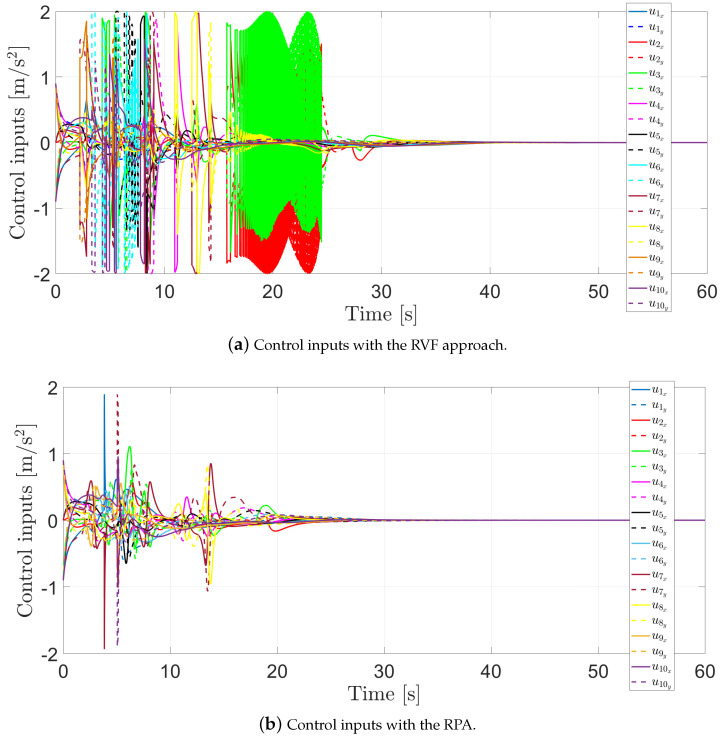
Control inputs.

## Data Availability

Not applicable.

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
