# Peer review of "Formation with Non-Collision Control Strategies for Second-Order Multi-Agent Systems"

_entropy, 2023, doi:10.3390/e25060904_

Round 1

Reviewer 1 Report

Dear authors, 

This paper proposes an interesting method for non-collision control strategies. However, I consider that there are some unclear points needing a revision. I list my concerns as follows:

Mayor issues:

- Despite presenting a novel technique, it is difficult to see how well your improvement performs without comparison to other algorithms. I recommend that you take a classic and well-known algorithm and make a comparison between the two.

If you don't compare with a traditional algorithm, you can't see if your improvements work better than others. You mentioned in the introduction, that you have some articles that talk about other methods, so you have the ability to improve the article by making a comparison with one of each development.

Apart from that, I consider that the article is well presented, the figures are also correct.

Reviewer 2 Report

The paper proposes a mathematical model for control of agents like mobile robots with inertia that can not change their velocity immediately. The idea is to detect that the distance to other robots is becoming small enough and at that point to apply appropriate control inputs to ensure that the distance is not smaller than safety distance. The paper includes all the necessary details of the mathematical model and the model has been validated by numerical simulations within two scenarios. In both cases considered in the article the agents reach their final destinations and all the time the distances of the robots remain sufficcent. Still, it is not clear if the model will actually work in other scenarios. Also some trajectories in Figure 6 do not look natural, for example the dark blue line. Maybe they can be explained?

Also it would be interesting to see how it works on physical robots that have sensor errors. This has not been done yet.

It should be possible to identify the particular agents whose parameters are shown in positioning charts (for example Figure 7). Otherwise it is not possible to match which agent from Figure 6 corresponds to which line if Figure 7. Maybe the same labels can be used for the same agent in both figures.

Careful proofreading is mandatory, preferably by native English speaker, especially the beginning of the paper should be improved. The mathematical and experimental parts are well written.

Almost 1/3 of references are self citations. The reviewer is not an expert in collision avoidance literature, but still has a strong belief that more works by other authors should be analysed. For example, what mathematical models are used by other authors?

Reviewer 3 Report

This paper proposes an approach to formation control of second order linear systems. Remarks:

-        - The English of the paper needs considerable improvements. Some formulations even compromise the readability of the study. E.g. in the Abstract “a parameter … is developed” or line 120 “will be designed taken into account”,  ….

-        - “Remark 1. In this work, directed, undirected and mixed communication topologies are taken into account” All of them? Simultaneously? Much clear formulation is necessary.

-       -  A more extensive literature survey related to the formation control of mechanical systems should be performed. More general results are available to formation control of Euler-Lagrange systems that consider collision avoidance. The authors should clearly emphasise the advantage of their proposed method related to the previous results presented in the literature.

-       -  Similar notions as “detection distance” and “security distance” were already applied in previous works related to formation control, they can hardly be considered new results.

-      -   The authors claim that one of the original contribution of this paper is that they solve the formation control problem by applying bounded control action. However, it seems that during collision avoidance the boundedness of the control signals are not considered. The simulation results also show high jumps in the control during collision avoidance.

-        -The convergence of the formation tracking error, and the collision avoidance are treated in separate theorems. How the formation error converges during collision avoidance?

-        -In my view the security distance “d” should be considered during the design of “c_ij” (desired relative positions) for proper convergence.  

-        -Eq. 8 (and the corresponding eqs. after it, e.g. the definition \xi_1): instead of “\tilde z + v” use “\lambda * \tilde z + v” where \lambda >0. You cannot directly add a position value (m) to a velocity value (m/s). Please refer e.g. to PD control or sliding mode control of mechanical systems.  

-        -Sentence after eq. (20) please add “Integrating (20) with integration limits t_0 and t”

-       -Line 38: please correct “some works some works”

Reviewer 4 Report

This article addresses the problem of formation control with non-collision for a multi-agent system with second-order dynamics. However, the study is lacking in related research analysis. Obstacle avoidance is a well-established research area, and therefore, a thorough analysis of related work in this area is required.

Moreover, while this study considered both formation and obstacle avoidance simultaneously, it failed to demonstrate the advantages of each. In the case of formation control, it is important to show that the formation is maintained well over time, which was not adequately demonstrated in this study. Similarly, the method used for obstacle avoidance, RVF, is a dated approach that does not reflect the latest research in the field.

Furthermore, the experimental results presented in the study are subjective, and a more objective evaluation using benchmarking simulations or actual experiments is necessary. Additionally, the study should describe the differences between the proposed method and conventional methods in the field.

Round 2

Author Response

We thank the reviewer for his valuable comments.

Reviewer 3 Report

The authors have addressed most of my questions.

 I have only one more request related to the answer to question 5.

From a mathematical point of view, one can obtain the following bound from the RVFs as follows           âˆ¥βi∥ 2Dϵρ    (1)”.

I wasn’t able to find this equation (1) in the paper. Please present this relation in the paper and the way it was obtained.

Reviewer 4 Report

The authors made sufficient revisions based on the previous review comments. However, I am still uncertain about the appropriateness of the selected conventional repulsive method, as it seems outdated. The authors should demonstrate the contribution and performance of their study adequately by comparing it with recent methods.
